# Neuroprotective Function of Rasagiline and Selegiline, Inhibitors of Type B Monoamine Oxidase, and Role of Monoamine Oxidases in Synucleinopathies

**DOI:** 10.3390/ijms231911059

**Published:** 2022-09-21

**Authors:** Makoto Naoi, Wakako Maruyama, Masayo Shamoto-Nagai

**Affiliations:** Department of Health and Nutrition, Faculty of Psychological and Physical Science, Aichi Gakuin University, 12 Araike, Iwasaki-cho, Nisshin 320-0195, Aichi, Japan

**Keywords:** Parkinson’s disease, synucleinopathy, monoamine oxidase, rasagiline, selegiline, neuroprotection, mitochondria, gene induction, α-synuclein, multiple-targeted drugs

## Abstract

Synucleinopathies are a group of neurodegenerative disorders caused by the accumulation of toxic species of α-synuclein. The common clinical features are chronic progressive decline of motor, cognitive, behavioral, and autonomic functions. They include Parkinson’s disease, dementia with Lewy body, and multiple system atrophy. Their etiology has not been clarified and multiple pathogenic factors include oxidative stress, mitochondrial dysfunction, impaired protein degradation systems, and neuroinflammation. Current available therapy cannot prevent progressive neurodegeneration and “disease-modifying or neuroprotective” therapy has been proposed. This paper presents the molecular mechanisms of neuroprotection by the inhibitors of type B monoamine oxidase, rasagiline and selegiline. They prevent mitochondrial apoptosis, induce anti-apoptotic Bcl-2 protein family, and pro-survival brain- and glial cell line-derived neurotrophic factors. They also prevent toxic oligomerization and aggregation of α-synuclein. Monoamine oxidase is involved in neurodegeneration and neuroprotection, independently of the catalytic activity. Type A monoamine oxidases mediates rasagiline-activated signaling pathways to induce neuroprotective genes in neuronal cells. Multi-targeting propargylamine derivatives have been developed for therapy in various neurodegenerative diseases. Preclinical studies have presented neuroprotection of rasagiline and selegiline, but beneficial effects have been scarcely presented. Strategy to improve clinical trials is discussed to achieve disease-modification in synucleinopathies.

## 1. Introduction

In the 1950s, iproniazid (*N*-isopropylisonicotinohydrazine) used for treatment of tuberculosis was found to show mood-elevating properties, which made impact on development of present neuro-psychopharmacology. Iproniazid was found to inhibit monoamine oxidase (MAO; EC1.4.3.4) and the finding inspired research on the role of monoamine neurotransmitters, MAO and MAO inhibitors, in neuropsychiatric disorders [1]. MAO was classified into type A and B (MAO-A, MAO-B) according to the substrate specificity and sensitively to inhibitors. The first generation of antidepressants, tranylcypromine and phenelzine, inhibit MAO-A, which is the major MAO in gastroentero-system and oxidizes tyramine, noradrenaline (NA), and adrenaline (A). The MAO-A inhibitors inhibit the enzymatic breakdown of tyramine contained in foods (cheese, red wine), increase release of NA and A and cause serious side effects of hypertension crisis, called “cheese effects”. In search for selective MAO inhibitors without “cheese effect”, Knoll et al. synthesized selegiline [(-)deprenyl, (*R*)-*N*,2-dimethyl-*N*-2-propynyl-phenylethylamine], a selective MAO-B inhibitor [2], but production of amphetamine a metabolite caused harmful sides. Later, rasagiline [*N*-propargyl-(1*R*)(+)aminoindan] was synthesized as a potent selective MAO-B inhibitor without amphetamine-like activity [3].

Synucleinopathies are a heterogeneous group of neurodegenerative diseases and share common pathological lesions with accumulation of misfolded insoluble α-synuclein (αSyn). They include Parkinson’s disease (PD), dementia with Lewy bodies (DLB), and multiple system atrophy (MSA). These synucleinopathies show different clinical features, etiopathology, histopathological findings, and genetic background [4]. αSyn is a protein of 140 amino acids, highly expressed at the presynaptic compartment, which regulates synaptic maintenance, dopamine (DA) neurotransmission, mitochondrial homeostasis, and proteasome function. Under physiological conditions, αSyn exists as a soluble unfolded monomer or tetramer in the cytoplasm and αSyn aggregates are present in the brain of elderly individuals without neurological disorders at low levels. Mutations of *SNCA* coding αSyn, environmental factors, and post-translational modification destroy the native conformation and induce misfolding and aggregation. αSyn is accumulated in Lewy bodies (LBs) in neuronal cell body, Lewy neurites (LNs) in the axon of idiopathic PD and DLB, and in glial cytoplasmic inclusions (GCIs) in oligodendroglia of MSA. In PD with minor cognitive decline, LBs and LNs are detected in DA neurons of the substantia nigra (SN) and in DLB a much larger number of LBs are present in the entorhinal and cingulate cortices. LB pathology is sometimes associated with Alzheimer’s disease (AD) with Parkinsonism, Down’s syndrome, various lysosomal storage diseases, such as Gaucher disease, and brain iron accumulation type-1 [5].

PD is the second common ageing-related neurodegenerative disease after AD. The primary pathology of PD is the selective degeneration of dopaminergic neurons in the SN pars compacta and deficit of DA content in the caudate putamen causes the onset of motor symptoms. At present, the etiology of sporadic form of PD is not established. From the clinical, epidemiological, molecular, and genetic aspects, PD is now considered as a large cluster of diseases resulted from a complicated interplay between genetic and environmental factors [6]. In family forms of PD, mutations in genes coding Parkin, PTE*N*-induced putative kinase 1 (PINK1), DJ-1, leucine-rich repeat kinase 2 (LRRK2), and αSyn have been identified as the etiologic factors. The pathogenic factors for neuronal loss include oxidative stress, mitochondrial dysfunction, deficits of neurotrophic factors (NTFs), especially brain-derived neurotrophic factor (BDNF), and glial cell line-derived neurotrophic factor (GDNF), neuroinflammation and accumulation of misfold αSyn [7]. Current DA replacement therapy using L-DOPA, DA receptor agonists, and MAO-B inhibitors can ameliorate symptoms. However, disease progression cannot be prevented or delayed. Neuronal loss advances in the multiple central and peripheral areas: NA neurons in the locus coeruleus, cholinergic neurons in the nucleus basalis of Meynert, serotonergic neurons in the median raphe, and further in the diverse brain regions and peripheral autonomic nerves.

Presence of αSyn in LBs and LNs, and the point and missense mutation and duplication of gene coding αSyn (*SNCA*) in early onset familiar PD, indicate its association with PD pathogenesis [8]. Braak proposed the neuropathological staging protocol for PD based on processes of αSyn accumulation [9]. αSyn accumulation begins in the gut and progresses to the brain, starts in the lower brain stem, the dorsal motor nucleus of glossopharyngyeal and vagal nerves, and anterior olfactory nucleus, and progresses in the limbic and neocortical regions at later stage. In PD and other synucleinopathies the disease progresses in prevalence to the motor phase, manifesting olfactory and autonomic dysfunction (gastrointestinal, urinary symptoms), rapid eye movement (REM) behavior, anxiety and depressive symptoms [10]. However, the ascending pattern of αSyn accumulation in the brain does not always apply to all synucleinopathies [11]. In DLB, parkinsonism appears after cognitive decline, whereas the reverse occurs in PD. The human brain has potential for plasticity and functional compensation, which might account for the latency of motor symptoms until about 70% DA neurons in the SN are lost. Intervention to disease progression during the prodromal stage is proposed as timing to start disease-modifying therapy and the prodromal signs may be applicable as biomarkers of the therapy efficacy [12].

Multiple risk factors in various neuronal populations are targeted to prevent neuronal loss for the disease-modifying therapy in PD [13]. Several clinical trials have been reported by use of antioxidants (vitamin E, *N*-acetylcysteine, inosine), nutrients (vitamin E, CoQ10, creatine), anti-inflammatory compounds (minocycline, erythropoietin), and GDNF, the DA neuron specific NTF, but most of the trials have not been able to get positive effects [14]. Selegiline was found for the first time to increase life expectancy and proposed to have the neuroprotective potency [15]. Selegiline and rasagiline are the most promising candidates of neuroprotective agents [16,17,18,19]. MAO-B oxidized *N*-methyl-4-phenyl-1,2,3,6-tetrahydropyridine (MPTP) into toxic *N*-methyl-4-phenylpyridinium ion (MPP^+^) and MAO-B inhibitors could prevent the toxicity in vivo and in vitro [20]. The DATATOP (Deprenyl and tocophenol anti-oxidative therapy for parkinsonism) study was the first double-blind placebo-controlled trial for the disease-modifying therapy of PD using selegiline and α-tocophenol [21]. Selegiline presented mild symptomatic benefit but could not modify disease progression [22]. A controlled, randomized trial of rasagiline, ADAGIO (Attenuation of Disease Progression with Azilert (rasagiline) Given Once-daily) was designed to evaluate the effects on disease progression by comparison of change in the Unified Parkinson’s Disease Rating Scale (UPDRS) score between early start and delayed start group. The early start group treated with rasagiline presented superior disease-modifying effects to the delayed start group [23,24]. A similar clinical trial with pramipexole PROUD (Pramipexole on Underlying Disease) in PD was also reported to show beneficial effects, but no neuroprotection was confirmed [25]. Delayed start trial of L-DOPA the LEAP (Levodopa in Early Parkinson Disease) could not confirm intervention into disease progression [26].

This paper reviews the molecular mechanism underlying neuroprotection by selegiline and rasagiline and the role of MAO in the pathogenesis in PD and neuroprotection by MAO-B inhibitors. Rasagiline and selegiline suppress oxidative stress, maintain mitochondrial function and homeostasis, and intervene mitochondrial apoptosis system. Selegiline and rasagiline prevent αSyn oligomerization and aggregation, promote the degradation, and mitigate the neurotoxicity. These MAO-B inhibitors enhance expression of anti-apoptotic Bcl-2 protein family and pro-survival NTFs. MAO-A was found to be associated with the gene induction by rasagiline. Multifunctional propargyl derivatives have been developed for neuroprotection for AD and other neurodegenerative diseases. The present state of clinical trials of “disease-modifying therapy” by rasagiline and selegiline is reviewed. In contrast to positive results in preclinical studies, most clinical trials have not fully presented beneficial results, and problems in clinical trials are discussed to develop novel strategy for future neuroprotective therapy.

## 2. Is Monoamine Oxidase a Pathogenic Factor for Parkinson’s Disease?

Discrepant localization of MAO-A in catecholaminergic neurons and MAO-B in glial cells had been an enigmatic issue for the role of MAO in neuroprotective function of MAO-B inhibitors. Is MAO-B the primary causal factor in PD and MAO-A a bystander? Role of MAO-A and MAO-B in neuronal loss is discussed.

MAO-A and -B are two independent proteins encoded by separate genes localized in the X-chromosome (Xp11.23) in opposite direction with tail-to-tail orientation. MAO-A and MAO-B genes have identical intron and exon organization derived from duplication of a common ancestral gene millions of years ago [27]. MAOs are of 70% sequence identity and localized at the outer mitochondrial membrane (OMM). They catalyze the oxidative deamination of primary and some secondary amines according to the reaction equation:
RCH_2_NHR′ + H_2_O + O_2_ = RCHO + R′NH_2_ + H_2_O_2_



The aldehyde is metabolized by aldehyde dehydrogenase and aldehyde reductase into glycols and carboxylic acid. H_2_O_2_ is converted into hydroxyl radical (OH^.−^) one of potent reactive oxygen species (ROS) in the presence of iron by the Fenton rection.

MAO-A preferentially oxidizes serotonin (5-hydroxtyrptamine, 5-HT) and NA and is inhibited by clorgyline [3-(2,4-dichlorophenoxy)-*N*-methyl-*N*-prop-2-ynylpropan-1-amine], whereas MAO-B oxidizes β-phenylethylamine and benzylamine. DA is oxidized by both MAO isoenzymes, but mainly by glial MAO-B in the human SN. MAO-A is mainly expressed in catecholaminergic neurons and most abundant in the locus coeruleus, paraventricular thalamus, and the striatal terminalis. MAO-B is expressed in serotonergic and histaminergic neurons, astrocytes and ventricular cells and abundant in the ependyma, circumventricular organs, olfactory nerve layer, and ventromedial hypothalamus [28]. MAO-A level in the brain can already be determined before birth [29], contributing to embryonic development of brain architecture and affect postnatal behavior and emotional state [30,31,32,33]. MAO-B appears only at the postnatal stage and increases with ageing 7.1 ± 1.3% every 10 years and decreases monoamine levels in the ageing brain [34]. MAO-B contributes more than 80% of MAO in the human brain, suggesting that MAO-B may contribute mainly to neuronal loss in ageing-related neurodegenerative diseases. Figure 1 summarizes the role of MAO-A and MAO-B in PD pathology and neuroprotection by MAO-B inhibitors.

MAO-B oxidizes DA, increases oxidative stress, produces neurotoxins from protoxicants, causes astrogliosis, and induces neuronal loss of the parkinsonian brain. Single-nucleotide polymorphism (SNPs) A644G (rs1799836) in the intron 13 of *mao-b* is known to be functional polymorphism and increases MAO-B mRNA and activity [35,36,37]. However, the association of *mao-b* 1799836 variant with PD phenotype has been not fully established [38].

In human brain, MAO-B was not present in dopaminergic neurons of the SN pars compacta, but in neurons of the nucleus raphe dorsalis, the ventral tegmental area, and the medial reticular formation. The density of MAO-B-positive glial cells was correlated to neuronal loss in PD, suggesting that MAO-B increases by massive gliosis outside the SN and may contribute to vulnerability in PD [39,40]. MAO levels in DA deficiency disorders were investigated by quantitative immunoblotting assay in autopsied brain homogenates [41]. In PD, MAO-B increased in the frontal cortex (+33%) in consistent with increased astroglial markers, such as glial fibrillary acidic protein (GFAP) aggregates and heat shock protein-27 (Hsp2) fragments but did not change in the SN. In MSA, MAO-B levels increased in the degenerating putamen (+83%) and SN (+10%), which might be associated with loss of MAO-B-rich serotonergic neurons. In progressive supranuclear palsy (PSP), MAO-B increased in the caudate (+26%), putamen (+27%), frontal cortex (+31%), and the SN (+23%). MAO-A levels increased in the PD putamen (+29%) and the PSP caudate (+19%), whereas it decreased in the MSA putamen (−23%). Levels of MAO-A 25 fragment were significantly increased in the PD SN (+33%), MSA (+40%), and PSP (32%), suggesting the possible association of MAO-A with deregulation of DA system in synucleinopathies. In induced pluripotent stem cells (iPSCs)-derived DA neurons prepared from patients with triplicated *SNCA* gene, MAO-A expression increased [42], MAO-A, and MAO-B in *Parkin* mutation [43] and MAO-B in *LRRK2* mutation [44]. Parkin degraded estrogen-related receptors and inhibited the expression of MAO-A and MAO–B, whereas loss of Parkin function increased MAO-A, DA oxidation, and ROS generation [45,46].

MAO-A mediates neurotoxicity of an endogenous dopaminergic neurotoxin *N*-methyl(*R*)salsolinol and mitochondrial toxins (rotenone, malonate and staurosporine) in cellular models [47,48,49,50]. MAO-A knockdown with small-interfering RNA or microRNA prevented binding to mitochondria and protect cells from apoptosis. MAO-B was overexpressed in SH-SY5Y cells but the sensitivity to *N*-methyl(*R*)salsolinol was not affected [48]. MAO-A regulates apoptotic signal pathway at downstream of p38 kinase and Bcl-2 and upstream of cell proliferation pathway, cyclinD1/E2F1 [51]. MAO-A increases ROS generation in mitochondria, induces p53 accumulation and DNA damage, inhibits parkin-mediated mitochondrial autophagy, and exerts cytotoxicity in mouse senescent cardiomyocytes [52]. Selegiline inhibited MAO-B activity totally in human brain, but MAO-A activity was still evident, suggesting that MAO-A may partially contribute the in vivo effects of selegiline [53]. Association of MAO-A with neuroprotection by rasagiline is discussed later [54,55].

Both MAO-A and -B are associated with oxidative stress and neuronal loss in the brain. The postmortem investigation of the human brains presents MAO-A increase in the SN of PD and MSA, whereas MAO-B decreased in PD and increased in MAS mainly by gliosis. MAO-A mediates signaling pathways to cell death caused by neurotoxins.

## 3. Pharmacological Properties of Selegiline and Rasagiline

Selegiline and rasagiline have similar chemicals structures and the same enzymological mechanism to inhibit MAO-B activity. However, the pharmaco-kinetical properties and in vivo effects are quite different.

Selegiline and rasagiline are potent and highly selective inhibitors of MAO-B with a propargyl residue attached to the amino group of methamphetamines or of aminoindan. The chemical structures of selegiline, rasagiline, and their metabolites are shown in Figure 2. Selegiline is the first MAO-B inhibitor applied for the treatment of PD as monotherapy and an adjunct of L-DOPA [56]. Selegiline and rasagiline bind to the active site of MAO, form a covalent adduct with the cysteinyl residue of the FAD cofactor by interaction between the *N*-propargyl group and *N*-5 of FAD isoalloxazine and inhibit MAO irreversibly [57]. Rasagiline inhibits MAO-B in the brain more potentially and more selectively than selegiline [16]. In the human brain, MAO-A and MAO-B are quantified using position emission tomography (PET) and radiotracer [58]. The half-life for de novo synthesis of MAO-B is 30–40 days significantly longer than that for MAO-A (2 days), suggesting that inhibition of MAO-B activity remains longer than that of MAO-A.

Oral administered selegiline is rapidly metabolized in the liver by CYP2B6 and a lesser extent by CYP2A6 and CYP3A4 into L-methamphetamine, which is further metabolized to L-amphetamine and desmethylselegiline [59]. L-Amphetamine and related metabolites cause adverse effects in the brain and cardiovascular system, such as appetite suppression and insomnia. On the other hand, rasagiline is metabolized by CYP1A2 into 1(*R*)-aminoindan, which has neuroprotective potency [60].

Selegiline and rasagiline show marked differences in the pharmacokinetic properties, like absorption, distribution volume (V_D_), metabolism, and elimination. Selegiline (10 mg/day) has a lower bioavailability than rasagiline (1 or 2 mg/day) (Table 1).

Selegiline and rasagiline are metabolized into either a toxic or a protective compound, respectively, which determines the quite different pharmacological effects in the brain.

## 4. Oxidative Stress and Mitochondrial Dysfunction in Synucleinopathies

Mitochondrial dysfunction enhances ROS production and is one of the major pathogenic factors for PD. Complex I deficiency in the mitochondrial electron transport chain (ETC) was once proposed as the typical pathological feature and etiological factor of PD, but this hypothesis is now not generally accepted.

Oxidative stress is caused by imbalance between excess production of ROS and reactive nitrogen species (RNS) and decreased activity of cellular redox system. In PD, DA metabolism, high content of iron and calcium (Ca^2+^) in the SN, mitochondrial dysfunction, and neuroinflammation cause oxidative stress [63,64,65]. DA autoxidation produces superoxide (O_2_^.−^) and toxic DA quinone, whereas the enzymatic oxidation by MAO produces 3,4-dihydroxyphenylacetaldehyde (DOPAL), hydrogen peroxide, and hydroxyl radical [66]. Under physiological state oxidative phosphorylation in the ETC produces O_2_^.−^ in complex I (NADH dehydrogenase) and to the less extent in complex III (cytochrome bc1 complex) and IV (cytochrome c oxidase). In neuronal cells, NADPH oxidase, xanthine oxidase, and endothelial nitric oxide synthase (eNOS) generate ROS and RNS. In the SN of the parkinsonian brains, enhanced oxidative stress is indicated by increased levels of lipid peroxidation (lipid hydroperoxides, 4-hydroxynonenal, advanced glycation end product), protein oxidation (protein carbonyls), and DNA oxidation (8-hydroxyguanosine) [67]. In the SN of the control aged similar pathological features are detected but not in the caudate nucleus, suggesting that neurons in the SN are continuously exposed to intensive oxidative environment [68,69]. Mitochondria are vulnerable to oxidative damage. Increased ROS/RNS damage mitochondrial DNA (mtDNA), oxidize the arginine, lysine, threonine, and proline residues in mitochondrial proteins, the ETC, mitochondrial carriers associated with apoptosis machinery and lipids [70]. In PD animal models, MPTP, rotenone, paraquat, *N*-methyl(*R*)salsolinol, manab (manganese ethylene-bis-dithiocarbamate), fenazaquin [4-(4-(tert-butyl)-phenethoxy)-quinazooline], and iron inhibit mitochondrial function, increase ROS/RNS production, and degenerate nigral DA neurons [71,72].

Deficits of complexes I and IV were detected in the SN, platelet, and muscle of parkinsonian patients [73,74]. Declined mitochondrial function in fibroblasts was correlated with clinical severity in idiopathic PD [75]. Dysfunction of the ETC was proposed as the hallmark of PD pathology, but it is not certain if the ETC deficit may be primarily genetic or secondary to accumulation of αSyn in mitochondria and its role in PD pathogenesis has been not fully established [76,77]. Only 30% of parkinsonian patients have clear complex I deficit, suggesting the occurrence of complex I deficiency in a distinct PD subgroup [78]. Exposure to environmental toxins interfering the ETC has been proposed to increase MSA risk [79]. 3-Nitropropionic acid (3NP), an environmental Complex II inhibitor, induced MSA-like symptoms in animals and αSyn-overexpressed mice [80,81]. Mitochondrial dysfunction was detected in fibroblasts of MSA patients, which might be caused by coenzyme Q deficiency [82].

Mutation and polymorphisms of familiar PD-responsible genes, *DJ-1*, *PINK1*, *PRKN*, *SNCA*, and *LRRK2*, are detected in the sporadic PD and sensitize to apoptosis signaling in cellular models [83]. Mitochondrial serine/threonine kinase *PINK1* mutations [84] and the E3 ligase *Parkin* loss [85] induce oxidative stress and mitochondrial dysfunction in the cellular and animal models. Loss of oxidative stress sensor DJ-1 impairs the ETC activity, increases ROS, reduces mitochondrial transmembrane potential (ΔΨm), and mitochondria are fragmented [86].

Mitochondria are deregulated by environmental toxins and genetic factors in PD and MSA. ETC impairment may not be the primary genetic factor in PD. Deregulated mitochondria cause not only oxidative stress, ATP depletion, but also activate death signaling as discussed below.

## 5. Selegiline and Rasagiline Modulate Oxidative Stress, Apoptosis Machinery and Homeostasis of Mitochondria

Selegiline and rasagiline exert neuroprotection in cellular and animal models of neurodegenerative disorders. Their neuroprotective functions include antioxidant function, direct regulation of apoptosis machinery in mitochondria, maintenance of mitochondrial biogenesis (mitogenesis), dynamics (fission, fusion), and autophagy (mitophagy). Recent results are reviewed.

In cellular and animal models, selegiline and rasagiline protect neurons from cytotoxicity of dopaminergic [MPTP, 6-hydroxydopamine (6-OHDA), *N*-methyl(*R*)salsolinol], noradrenergic [*N*-(2-chlorothyl)-*N*-ethyl-2-bromo-benzyl-amine hydrochloride (DSP-4)], serotonergic [5,6-dihydroxyserotonin], cholinergic [ethylcholine aziridinium (AF64A)], and excitotoxic (3NP) toxins, oxidative stress, NTF depletion, ischemia, and axotomy [87,88,89,90]. Neuroprotection of selegiline and rasagiline is attributed to regulation of apoptosis system in mitochondria [91,92]. Selegiline and rasagiline also activate intracellular signaling pathways to induce anti-apoptotic Bcl-2 protein family and BDNF, GDNF, and antioxidant enzymes, and have anti-inflammatory functions [93,94,95,96].

### 5.1. Antioxidant Functions of Selegiline and Rasagiline

Enzymatic oxidation of tyramine, a MAO-A/B substrate, increased intra-mitochondrial H_2_O_2_ levels by 48-folds from the basal level and damaged mtDNA, suggesting that inhibition of MAO activity directly prevents oxidative stress [97]. Selegiline protected cellular models of PD induced by oxidative stress [peroxynitrite (ONOO^−^), nitric oxide (NO)] [98]. Propargylamine residue can directly scavenge NO and ONOO^−^, as shown by 1-phenylpropargylamine and *N*-propargyl tetrahydroisoquinoline [99,100]. Propargylamine is composed of an amine group in β-position to an alkyne moiety and compounds with a carbon-carbon triple bond can behave as electrophilic substrates and as electron source in nucleophilic reactions. Selegiline inhibits NOS in brain mitochondria, potentiates mitochondrial cytochrome oxidase (complex IV) activity, and reduces ROS/RNS production [101,102]. Selegiline and rasagiline induce nuclear translation of nuclear factor erythroid 2-related factor (Nrf2), increase binding to the antioxidant response element (ARE), and enhance expression of antioxidant thioredoxin super families. Selegiline, rasagiline, and 1(*R*)-aminoindan activate tyrosine kinase (Trk) receptor/phosphatidylinostol-3-kinase (PI3K) signal pathway or protein kinase C (PKC) and increase activities of glutathione-dependent antioxidant enzymes (glutathione peroxidase, glutathione-S-transferase) and anti-peroxidative enzymes, catalase, and superoxide dismutase (SOD) [103,104,105].

Selegiline and rasagiline inhibit MAO and its propargylamine residue direct scavenge ROS/RNS. They activate intracellular signaling pathways and induce expression of antioxidant enzymes.

### 5.2. Rasagiline and Selegiline Directly Inhibit Mitochondrial Apoptosis Cascade

In cellular and animal models of PD, selegiline and rasagiline have been reported to prevent apoptosis. To clarify the cellular mechanism, we developed a cell model of apoptosis presenting sequential opening of pore in the IMM and OMM.

Apoptosis has been detected in the parkinsonian brain, as shown by increase of active caspase-3 and decrease of Fas-associated protein with a death domain (FADD)-immunoreactive DA neurons in the SN [106]. Intrinsic apoptosis is initiated with mitochondrial permeability transition (MPT), a sudden increase in mitochondrial membrane permeability, followed by opening of the non-specific mega-channel called the mitochondrial permeability transition pore (mPTP) [107]. Figure 3 shows apoptosis process induced by neurotoxins and ROS/RNS and target sites for rasagiline and selegiline to prevent cell death. The mPTP opening progresses sequentially. Mild stimuli induce pore formation at the IMM in a reversible way and allow entry of water, solute with molecular mass less than 980 Da, metabolites, and inorganic ions into the matrix. More intense and prolonged insults further increase the membrane permeability in the OMM, open the mPTP expanding from the OMM, IMM to the contact site, release molecules with molecular mass up to 1500 Da and Ca^2+^, induce expansion of the matrix, and rupture of the OMM.

The mPTP opening causes ΔΨm collapse, releases apoptosis-executing molecules, cytochrome c (Cytc), second mitochondria-derived activator of caspase/direct IAP binding protein (Smac/Diablo), apoptosis-inducing factor (AIF), and endonuclease G, from the mitochondria to the cytoplasm, activates caspases, induces nuclear translocation of glyceraldehyde-3-phosphate dehydrogenase (GAPDH), condensation and fragmentation of nuclear DNA, and finally cell death. The major components of the mPTP are voltage-dependent anion channel (VDAC) at the OMM, adenine nucleotide translocator (ANT) at the IMM, and cyclophilin D (CypD) at the matrix, but the detailed structure has not been fully established. Anti- and pro-apoptotic Bcl-2 protein family, hexokinase I and II, are associated with VDAC on the cytoplasmic face of the OMM, and creatine kinases at the intermembrane space. Oxidative stress, ATP deficit and Ca^2+^-overload increase membrane permeability in mitochondria and activate apoptosis system, whereas cyclosporin A (CysA) inhibits CypD binding to ANT and prevents pore opening at the IMM. Antioxidant enzymes (catalase, SOD) and inhibitors of Ca^2+^ uniporter (ruthenium, *N*-ethylmaleimide, butacaine) suppress apoptosis. Oxidation of critical cysteine in ANT changes the conformation and opens the mPTP [108].

Rasagiline and selegiline prevent the mPTP opening and following apoptosis in cellular models [92,109,110,111]. A cell model was established by use of a TSPO ligand, PK11195 [1-(2-chlorophenyl)-*N*-methyl-*N*-(1-methylpropyl)-3-isoquinoline-carboxamide] [112]. After addition of PK11195, a pore opened at the IMM, and the burst of superoxide production in the ETC, called “superoxide flash” occurred. Then, the mPTP opening was observed by mitochondrial Ca^2+^ efflux into the cytoplasm. Rasagiline and selegiline prevented PK11195-induced mPTP formation and following apoptosis process. Rasagiline completely suppressed the initial pore formation at the IMM, superoxide flash followed by Ca^2+^ efflux, whereas selegiline prevented Ca^2+^, but did not prevent superoxide flash [113]. The propargylamine residue in rasagiline has a protonated amino group at the physiological pH, interacts with critical aromatic or amino acid residues in the ANT, and inhibits the complex formation of CypD with ANT at the IMM as in the case with CysA. On the other hand, selegiline prevented opening of the mPTP at the later phase of the mPTP opening as in the case with Bcl-2 overexpression. Selegiline and rasagiline enhance antioxidant enzymes, or directly scavenge ONOO^−^ to prevent oxidation of vital cysteine residues in the ANT and other mitochondrial protein.

Rasagiline directly regulates formation of the mPTP and prevents apoptosis in a cellular model. Selegiline inhibits apoptosis at the different stage of the mPTP formation.

### 5.3. Mitochondrial Biogenesis, Dynamics and Mitophagy in Neuroprotection

For cell survival, mitochondria should maintain their activity, homeostasis, shape, and number by regulation of biosynthesis and degradation of damaged mitochondria by autophagy. In PD, the dynamic processes of mitochondria are impaired. MAO and MAO-B inhibitors modulate mitochondrial recycling for neuroprotection.

Mitochondria are dynamic organelle undergoing continuous processes, biogenesis (mitogenesis), dynamics (fission, fusion), and autophagy (mitophagy) to maintain the function, structure, distribution, and movement. Mitogenesis dysfunction is one of the pathogenic factors in PD, AD, cardiovascular diseases, diabetes type II, and neuroinflammation [114]. Mitogenesis is regulated by AMP-activated protein kinase (AMPK)/sirtuin 1 (SIRT1) signaling pathway and peroxisome proliferator-activated receptor γ (PPARγ) cofactor-1α (PGC-1α) [115,116]. PGC-1α interacts with Nrf2, activates mitochondrial transcription factor A (TFAM), promotes the binding to promoter regions of nuclear genes encoding subunits of the five ETC complexes, and replicates mtDNA. Ca^2+^/CaMK (calmodulin-dependent protein kinase)/CREB [c-AMP response element (CRE) binding protein] pathways increase mitogenesis and show neuroprotection. In parkinsonian brain, PGC-1α and PGC-1α-targeted genes were downregulated, mitogenesis and mitochondria function were impaired, and PGC-1α was proposed as a target of neuroprotection [117,118]. Polymorphisms in the PGC-1α gene have been reported to be one of the risk factors of PD onset [119].

Mitochondria undergo frequent and rapid fusion and fission events in response to cellular demands and environmental stimuli to maintain mitochondrial quality. Defects in mitochondrial dynamics impair mitochondrial respiration, morphology, and mobility, and are involved in the pathogenesis of PD, AD, and Huntington’s disease (HD) [120,121]. Mitochondrial dynamics are regulated by fission proteins, dynamin-related protein-1 (Drp1) and fusion proteins, mitofusin-1 and -2 (Mfn1, Mfn2) at the OMM and optic atrophy protein (Opa1) at the IMM. PD-linked mutations of genes, *PINK1*, *parkin*, *DJ-1*, *LRRK2*, and vacuolar protein sorting-associated protein 35 (*VPS35*) affect genes encoding proteins involved in mitochondrial dynamics. Wild type LRRK2 interacts with Drp1, increases its phosphorylation and activates fission, whereas overexpression of wild or mutated *LRRK2* (PD-associated G2019S) increases mitochondrial fragmentation and mitophagy and causes apoptosis [122].

Autophagy degrades unwanted cytoplasmic aggregates and damaged organelles in a lysosome-dependent way and processes in sequential stages, fission, priming, engulfment and cleaving. There are 3 types of autophagy: macroautophagy (MA, usually called as autophagy), microautophagy, and chaperone-mediated autophagy (CMA). In mitophagy, mitochondria are degraded through binding to receptors or adaptors containing the microtubule-associated protein 1 light chain 3 (LC3)-interacting region, BNIP3 (BCL/adenovirus E1B 19kDa interacting protein 3), and Nip3-like protein X (NIX). Mitophagy plays an important role in preservation of intracellular energy, controls mitochondrial quality, appropriate the mass and population, and promotes cell survival in ageing and neurodegenerative diseases [123,124]. PINK1 and Parkin maintain healthy mitochondrial population. PINK1 is imported into mitochondrial matrix and is degraded by mitochondrial protease, presenilin-associated rhomboid-like protein (PARL). The cleavage of PINK1 activates Parkin, which polyubiquitinates Mfns 1/2 and Drp-1, induces their association with the ubiquitin-binding domains of autophagy receptors, and forms the autophagosomes. PINK1 and parkin mutation causes accumulation of damaged mitochondria in axon and within autophagosomes in neurons of PD/DLB brains [125]. Bcl-2 family regulates mitochondrial dynamics and autophagy through its binding to AMBRA1 (activating molecule in beclin 1-reglated autophagy), interaction with beclin-1 containing a BH_3_ domain, and inhibition of Parkin translocation to mitochondria [126].

MAO and its inhibitors are associated with protective and toxic effects of autophagy in the brain. In neurodegenerative disorders, excessive activation of autophagy results in neuronal cell death and accelerates the disease progress. MAO-A and MAO-B generate ROS by oxidation of monoamine substrates, inhibit Parkin-mediated mitophagy, and induce neuronal death [52,127]. On the other hand, MAO-A overexpression promoted protective autophagy by ubiquitination of mitochondrial protein, phosphorylation of Bcl-2 and dynamin-1-like protein, increased complex IV activity/protein, and mitochondrial function [128]. Small molecule compounds targeting the ALP have been proposed as the neuroprotective therapy through promoting autophagic degradation of pathogenic aggregates and damaged organelle [129]. Proteins associated with autophagy, such AMPA, mTOR, Beclin-1, TFEB, and ERRα, are the targets of metformin, rapamycin, related synthesized compounds, resveratrol, curcumin, and trehalose.

Antioxidant functions of rasagiline and selegiline can improve and protect mitochondrial function. Selegiline increased cytochrome oxidase activity and oxygen uptake in state 3 and prevented the MPT and cell death in mice [130]. Selegiline prevented increase of autophagy-related Beclin1 and LC3 in the SN of rotenone-induced rat model of PD and inhibited autophagy [131]. MPTP treatment upregulated expression of autophagic proteins, LC3, Beclin, ATG-5 and p62 in SH-SY5Y cells, and selegiline attenuated MPTP-induced autophagic response and protected cells [132]. Rasagiline prevented photoreceptor cell death in Prph2/eds mouse by induction of autophagy-related genes [133]. AMPK is a key factor in the regulation of autophagy and intracellular energy homeostasis. Rasagiline and 1(*R*)-aminoindan antagonize NMDA receptor, AMPA receptor, and metabotropic glutamate receptor and inhibit glutamate transmission, whereas selegiline attenuates excitability increase mediated by kainite receptor [134]. Rasagiline and selegiline suppress glutamate transmission in the hippocampus and indirectly modify mitogenesis in rats exposed to ischemia/perfusion [135]. Selegiline and rasagiline activate AMPK/CREB pathway and transcription factor 2 (ATF2) and induce PGC-1α transcription. PGC-1α increases Parkin and DJ-1 for neuronal survival [136] and prevents their inactivation by oxidative modification in the sporadic parkinsonian brain [137,138]. M30 [5-(*N*-methyl-*N*-propargyl-amino-methyl)-8-hydroxyquinone], a MAO inhibitor with iron-chelating activity, targets Sirt1/PGC-1α pathway and restores activity of DA neurons in animal PD models prepared with MPTP, lactocystin, and 6-OHDA [139].

MAO-A and -B activate PGC-1a and promote mitochondrial biosynthesis. They inhibit mitophagy by ROS-modified Parkin to neuronal death, whereas MAO-A overexpression promotes protective autophagy and increases mitochondrial function.

## 6. Synuclein the Common Pathogenic Factor in Synucleinopathies

Accumulation of αSyn aggregates is commonly observed pathological feature of synucleinopathies. Conformational changes of αSyn from monomer to oligomers are now considered to cause the accumulation and toxicity and proposed as the target site of neuroprotection.

αSyn is s one of the major targets of disease-modifying therapy in synucleinopathies [140,141]. Under physiological conditions, αSyn is attached to membrane of synaptic vesicles and presents in the cytoplasm as unfolded monomer or tetramer. Changes in the protein environments increase the hydrophobicity of αSyn, decrease the net charge, and induce folding. Aggregation pathways of αSyn from monomers to oligomers, protofibrils, and insoluble fibrils play a critical pathogenic role in synucleinopathy [142,143]. αSyn has three sequence regions: *N*-terminal region (NTR, residues 1–60), central hydrophobic region called non-amyloid β-component with a conserved motif (KTKEGV) (NAC, residue 61–65), and *C*-terminal region (CTR, residues 66–95). After binding to membrane, the NTR forms two α-helices and αSyn changes the conformation from a random coil to β-sheet structure and amyloid β-like fibrils containing β-sheet rich secondary structure. The CTR is highly acidic, anti-amyloidogenic, and involved in Ca^2+^ binding. The central hydrophobic region is the nucleation core, whereas the NTR and CTR are not directly associated with αSyn fibrillization. Many intrinsic and environmental factors, such as αSyn mutations, pH of the environment, and presence of chaperones, accelerate misfolding process of amyloidogenic protein. Aggregation is accelerated by the presence of fibrillar seed, cupper and iron, and posttranslational modification, such as binding of DA and DOPAL, phosphorylation, oxidation, glycosylation, methylation, ubiquitination, and modification by small ubiquitin-like modifier (SUMO) [144].

αSyn is trafficked into mitochondria by mitochondrial import receptors, translocases of the OMM (TOM) proteins, TOM70, TOM40, and TOM22, and anchored at the matrix and IMM [145]. In the brains of parkinsonian patients, αSyn is accumulated in mitochondria, TOM40 significantly reduces, mtDNA deletion, and oxidative DNA damage occur, and mitochondrial function is impaired [146]. Under physiological conditions, monomeric αSyn interacts with subunit of ATP synthase improves its efficacy [147]. β-Sheet-rich αSyn oligomers interacts with ATP synthase, oxidize its β-subunit, impair respiration, induce mitochondrial dysfunction, stimulates mitochondrial lipid peroxidation, open the mPTP, and induce apoptosis [148]. Oligomeric αSyn is imported into mitochondria also through VDAC at the OMM and inhibits complex I function [149]. It interacts with ANT at the IMM, increases ROS production, induces Ca^2+^ overload, ΔΨm decline, opens mPTP and apoptosis. Phosphorylated non-fibrillar αSyn is translocated into mitochondria and induces apoptosis [150]. Overexpression of αSyn A53T mutant in mice induced accumulation of αSyn monomers and oligomers in mitochondria, damaged membrane stability, inhibited complexes I and III, and increased mitochondrial autophagy in DA neurons [151,152,153]. αSyn overexpression activated NF-κB/ PI3K/Akt pathway, decreased anti-apoptotic Bcl-2, and enhanced apoptogenic Bax expression and induced apoptosis [154]. Under oxidative stress, nuclear αSyn is bound to PGC-1α promoter, the major transcription factor for mitogesis, inhibits mitochondrial function, and enhances the vulnerability of the nigral DA neurons to αSyn in mice [155,156].

In the PD brains, activation of glial cells, infiltration of lymphocytes, and presence of inflammasomes were detected, suggesting the involvement of inflammation in the onset and progression [157]. Infection of influenza A, Herpes simplex virus-1 (HSV-1), Ebola virus, Japanese encephalitis virus, and human immunodeficiency virus has been reported to cause parkinsonism [158]. Increased levels of inflammatory mediators, TNFα, nitric oxide (NO), PGF2, IL-1β, IL6, and oxidized molecules were found in the parkinsonian SN and cerebrospinal fluid (CSF) [159,160]. Mutations of *parkin*, *SNCA*, *PINK1*, *DJ-1*, and *LRRK2* activate reactive astrocytes, microglia and increase expression of monocytes and glial cells. PD is now proposed as an inflammatory disease stemming from pathogenic inflammation in the gut [161]. αSyn in the enteric nervous system has potent chemoattractant activity of neutrophils and monocytes in response to viral infection. Chronic infection in the gastrointestinal tract and breakdown of epithelial barrier cause αSyn accumulation and aggregation, and αSyn trafficking into the brain might be another way to induce PD.

Oligomeric αSyn is accumulated in mitochondria, deregulates the ETC, activates apoptogenic Bax, and induces apoptosis. αSyn localized in the enteric nervous system activates inflammatory reactions in PD.

## 7. Rasagiline and Selegiline Modify α-Synuclein Oligomerization and Aggregation

Rasagiline and selegiline bind to αSyn, change the conformation, turn it to non-fibril pathway and reduce the aggregate accumulation at the various stages. Selegiline and other anti-parkinsonian compounds destabilize preformed αSyn fibrils. αSyn and its degraded compound enhance expression of MAOs, which may promote loss of DA neurons.

αSyn processes to oligomers and aggregates and the distinct strains is implicated with the heterogenic pathology and phenotypes of synucleinopathies [162,163]. Neuroprotective phytochemicals, MAO-B inhibitors, and some anti-parkinsonian agents can intervene the fibril pathway and turn it to non-fibril pathway [164]. Rasagiline and 1(*R*)-aminoindan bind to αSyn, change the conformation to more loop structure and inhibit the linear structure required for the β-sheet formation, aggregation, and fibrilization [165]. Rasagiline forms a complex with αSyn more effectively with higher affinity than selegiline. Rasagiline binds to sites in the *N*- and *C*-terminals, whereas selegiline and methamphetamine only to the *N*-terminus. Rasagiline alters profiles of the NTC and CTR and induces folding between the *N*- and *C*-terminals, as in the case with DA and Cu^2+^ [166]. Methamphetamine and selegiline bind to the NTR, changes the conformation more compact without forming β-sheet, but do not prevent αSyn aggregation [167]. Selegiline interferes with αSyn nuclear formation, delays fibril formation of A30P and wild type αSyn, induces amorphous heterogeneous aggregates, and reduces cytotoxicity [168]. Selegiline and other anti-parkinsonian compounds destabilized preformed αSyn fibrils in the order of selegiline = DA > pergolide > bromocriptine but the mechanisms of these compounds were not presented [169]. MAO-B inhibitors prevent enzymatic oxidation of DA, increases autoxidation of DA into DA-quinone, and promotes non-fibril pathway of αSyn. Selegiline and rasagiline preferentially increased secretion of detergent-insoluble αSyn and decreased intracellular accumulation in vitro, which is mediated by the ATP-binding cassette (ABC) transporter, non-classical secretion pathway. In rats transfected with recombinant adeno-associated virus (rAAV)-A53T αSyn, selegiline delayed αSyn aggregation in the striatum and mitigated loss of DA neurons in the SN [170].

αSyn and the metabolites affect MAO activity and expression. αSyn selectively binds to MAO-B and enhances the activity [171]. Asparagine endopeptidase (AEP, legumain, δ-secretase) is a cysteine endopeptidase and is proposed to play an important role in PD and AD pathogenesis [172]. AEP leaves 1–103 fragment (N103) from αSyn, which binds intensively to MAO-B, but not MAO-A, enhances MAO-B activity and the neurotoxicity, and promotes PD pathology in vivo. In the SN of the parkinsonian brain, N103 level and AEP activity increased. DOPAL, 3,4-dihydroxyphenyl glycolaldehyde (DOPEGAL, a NA metabolite by MAO-A) and MPP^+^ activated AEP and increased N103 production in cellular models [173]. Rasagiline and MAO-B knockout could prevent the toxicity and inhibition of the AEP/MAO-B pathway, which might be another mechanism of the neuroprotection by rasagiline [172]. Overexpression of wild type and A53T αSyn upregulated MAO-A protein expression by selective induction of Sp1, a transcription factor of MAO-A, and promoted depressive-like behaviors in mice [174,175]. The impairment of MAO-A by αSyn may be associated with depression in PD.

Selegiline and rasagiline prevent αSyn fibril formation in vitro by different mechanisms. MAO-B expression is enhanced by αSyn and its metabolite, which rasagiline and selegiline can prevent. αSyn overexpression promotes MAO-A expression, suggesting that crosstalk between αSyn and MAO is involved in the symptoms of PD. AEP-catalyzed N103 fragment of αSyn should be further investigated in relation to the PD pathogenesis and neuroprotection by MAO-B inhibitors.

## 8. Selegiline and Rasagiline Induce Neuroprotective Bcl-2 and Neurotrophic Factors, Which MAO-A in Neurons Mediates

Selegiline and rasagiline have been shown to protect neurons independently on inhibition of MAO activity. They induce expression of anti-apoptotic Bcl-2 protein family and neuroprotective neurotrophic factors, especially GDNF and BDNF in cellular and animal models and also in patients with PD. Role of MAO-A in the gene induction by MAO-B inhibitors was indicated.

Anti-apoptotic Bcl-2, Bcl-xL, MC-1, and Bcl-w regulate apoptotic processes by multiple mechanisms: sequestration of the apoptosis-activator BH3-only protein, prevention of Bax activation, binding to the mPTP components, reducing mitochondrial oxidative stress, promoting proton efflux to maintain ΔΨm, increasing in ΔpH and intra-mitochondrial K^+^ and Ca^2+^ buffering, and downregulating the density of contact sites between the OMM and IMM [176]. Anti-apoptotic Bcl-2 levels in lymphocytes were negatively correlated with the duration and severity of PD. Treatment with L-DOPA and dopamine agonists downregulated Bcl-2 levels and increased the density of apoptosis-inducing TSPO, suggesting that the disease progression and therapy affect mitochondrial apoptosis system [177]. Bcl-2 decreased in MPTP-induced cellular and animal models of PD [178], but the direct involvement of Bcl-2 in PD pathogenesis remains speculative.

Rasagiline and selegiline upregulate the expression of anti-apoptotic Bcl-2, Bcl-xL, and Bcl-w and downregulate that of apoptogenic Bad and Bax in cellular and animal models [179,180,181]. MAO-A and MAO-B mediate the gene induction by selegiline and rasagiline in neural and glial cells differently, as summarized in Figure 4.

MAO-A knockdown with (*si*) RNA downregulated Bcl-2 induction by rasagiline but not by selegiline in neuronal SH-SY5Y cells [182]. In glial U118MG cells, MAO-B knockdown significantly increased the expression of Bcl-2, BDNF, and GDNF, and selegiline synergistically enhanced the gene induction, whereas MAO-B knockdown decreased the gene induction by rasagiline [183]. MAO-A knockdown inhibited the gene induction by selegiline and rasagiline in U118MG cells. Rasagiline activates PKC/extracellular-regulated kinase (ERK)/NF-κB signal pathways and induces the expression of Bcl-2 and BDNF. The propargylamine moiety is essentially required for gene induction by rasagiline, selegiline, TV3326 [(*N*-propargyl)-(3*R*)-aminoindan-5-yl]-ethylmethyl carbamate], aliphatic propargylamines, and *N*-propargylamine [184,185,186].

GDNF supports the viability of dopaminergic midbrain neurons in tissue culture [187]. GDNF regulates physiologically survival and death of DA neurons and promotes survival of midbrain DA neurons in animal PD models. GDNF binds to its receptor GFRα1 and activates receptor tyrosine kinase Ret and downstream signaling pathways [188]. In the SN of parkinsonian brain, GDNF protein most markedly decreased in neurons and neutrophil, followed by cilliary neurotrophic factor (CNTF) > BFNF, but NGF and neurotrophin-3 and -4 (NT-3, -4) did not decrease [189]. In the parkinsonian hippocampus, GDNF significantly decreased, whereas fibroblast growth factor 2 and cerebral dopamine neurotrophic factor markedly increased. BDNF and mesencephalic astrocyte-derived neurotrophic factor did not change [190]. BDNF is expressed specifically in the SN pars compacta and BDNF protein in melanized neurons of the SN decreased signify in PD, and BDNF deficit was proposed to be involved in the PD pathology [191]. BDNF activates TrkB/receptor-mediated MAPK/ERK, PI3K, and phospholipase C (PLC)-γ pathways and protects DA neurons against toxicity of MPTP or 6-OHDA [192]. Clinical trials with GDNF and neurturin by means of various delivery systems in the putamen of parkinsonian patients were reported, but the therapeutic benefit has not been fully approved [193,194].

Selegiline, rasagiline and 1(*R*)-aminoindan induce expression of GDNF, BDNF, NT-3, NGF, and the BDNF-receptor TrkB in cellular and animal models [93,94,195,196,197,198]. Selegiline treatment (5 mg/day for 2–8 weeks) significantly increased BDNF and GDNF levels in the CSF of parkinsonian patients [94]. Rasagiline was systematically administered to non-human primate for 4 weeks and rasagiline at 0.25 mg/day increased GDNF and at 0.25 and 0.1 mg/day BDNF, NGF, and NT-3 in the CSF. In SH-SY5Y cells, rasagiline (10^−7^–10^−10^ M) increased GDNF expression, whereas selegiline (10^−7^–10^−9^ M) enhanced BDNF significantly. Rasagiline and selegiline increase GDNF and BDNF at quite low concentrations and the definite concentrations are required to optimize the efficient gene induction.

Rasagiline and selegiline activate signaling pathways to promote expression of neuroprotective genes in clinical and preclinical trials. MAO-A in neuronal cells and MAO-B in glial cells are associated with gene induction by the MAO-B inhibitors. Rasagiline and selegiline enhance preferentially GDNF and BDNF, respectively.

## 9. Neuroprotective Multifunctional Propargylamine Derivatives Developed from Selegiline and Rasagiline

From the chemical structure of rasagiline containing a neuroprotective propargylamine residue, various multifunctional compounds have been developed. They were designed to target multiple pathogenic factors.

Multi-targeted drugs with MAO and cholinesterase (ChE) inhibition, metal chelating and antioxidant activity have been developed from rasagiline and selegiline for treatment of PD, AD, HD, and amyotrophic lateral sclerosis (ALS) [199,200,201]. Chemical structures of the major compounds are shown in Figure 5. A series of iron-chelating neuroprotective compounds were synthesized. M-30 5-(*N*-methyl-*N*-propargyl-aminomethyl)-8-hydroxyquinoline was developed from VK-28 5-[4-(2-hydroxyethyl)piperidine-yl-methyl]-quinoline-8-ol by introducing a neuroprotective propargylamine moiety into ion-chelating 8-hydroxyquinone (Figure 5A). 

M30 has high iron chelating and antioxidant activities, such as direct scavenge of hydroxyl radicals and inhibition of lipid peroxidation. M30 inhibits MAO-A and MAO-B in the brain in vivo and shows neuroprotective activity in animal and cell models of AD, ageing, PD, and ALS. M30 and HLA-20 5-[4-propargyl-piperizin-1-yl-methyl]-8-hydroxyquinoline improved memory loss in rat model of sporadic AD.

Hybrids of the carbamate ChE inhibitor rivastigmine and rasagiline, *N*-propargyl-(3*R*)aminoindan-5yl)-ethyl methyl carbamate (ladostigil, TV3326) and its *S*-isomer TV3279, were developed for treatment of dementia with Parkinsonism and AD [202] (Figure 5B). TV3226 inhibited MAO-B and ChE, exerted antiapoptotic activity much more potent than its *S*-enantiomer [185]. It is neuroprotective and inhibits acetyl- (AChE) and butyl-cholinesterase (BChE) and brain selective MAO-A and MAO-B. Ladostigil markedly decreased holo-amyloid precursor protein (APP) levels, in addition to neuroprotection. In a 3-year, randomized, double-blind, placebo-controlled phase 2 clinical trial in mild cognitive impairment (MCI), ladostigil was safe and tolerated, but could not delay progression to dementia [203]. A series of conjugates of a propargyl group to *N*-methyl position of rivastigmine were synthesized and *R*-enantiomer of MT-20 was the most potent inhibitor of MAO-B and ChE, increased Bcl-2, and downregulated Bax and caspase 3 expression in PD model mice [204]. MT-031 was synthesized by replacement of a methyl group at *N*-position of rivastigmine with a propargyl group, inhibited MAO-A and AChE/BuChE, enhanced DA, 5-HT and NA in the striatum, upregulated mRNA expression of Bcl-2, BDNF, GDNF and NGF, exerted anti-inflammatory and antioxidant activities in scopolamine-treated mice and cultured astrocyte [205]. A series of carbamate derivatives of 1,3-rasagiline have been synthesized and examined for neuroprotective potency by use of computational chemical techniques base on quantitative structure-property relationship (QSAR) [206]. A hybrid of donepezil, propargylamine, and 8-hydroxyquinone was synthesized, in order to inhibit MAOs and ChE and chelate metals. Racemic α-aminotrile-4-(1-benzylpiperdin-4-yl)-2-{[(8-hydroxyquinone-5-yl)methyl]pro-2-yn-1-yl}amino butaneitril (DPH6) was the most potent and ameliorated scopolamine-induced learning deficits in healthy mice [207]. ASS234, *N*-[{5-[3-(1-benzyloxypiperidin-4-yl)propxy]-1-methyl-1*H*-indol-2-yl}methyl]-*N*-methylprop-2-yn-1-amine was synthesized as another conjugate of the benzylpiperidine moiety of donebezil and the indolyl propargylamine moiety of a MAO-B inhibitor PF9601N, *N*-(2-propynyl)-2-(5-benzyloxy-indolyl) methylamine [208] (Figure 5C). It inhibited MAO-A more potently than MAO-B, AChE, and BChE moderately and was proposed as a candidate of potential neuroprotective drugs.

Indanone derivatives of rasagiline were found to be antagonists of histamine H_3_ receptor (H_3_R) and 5-substituted 3-piperidinopropyloxy-rasagiline was the most potent H_3_R antagonist and MAO-B inhibitor [209] (Figure 5D). Histamine H_3_ receptors (H_3_Rs) are distributed in the striatum, cortex and hippocampus, brain regions related to cognition, and regulate levels of histamine, DA, and acetylcholine in synaptic cleft. H_3_R antagonists are proposed as potential drugs for narcolepsy, AD, PD, and attention deficit hyperactive disorder (ADHD). Among a series of synthesized (hetero) arylalkenyl propargylamine compounds, SZV558, methyl-(2-phenyl-allyl)-prop-2-ynyl-amine, was a MAO-B inhibitor and antioxidant, protected dopaminergic cells in vitro against toxicity of rotenone, and prevented loss of DA neurons and motor dysfunction of mice treated with MPTP in vivo (Figure 5E) [210].

An increasing number of multi-functional propargylamine compounds are developed for treatment of various neurodegenerative disorders. The merit of multi-targeted compounds has not been established and requires further clinical trials in the future.

## 10. Clinical Trials of Selegiline and Rasagiline in Neuropsychiatric Disorders

As discussed above, preclinical studies have presented multiple neuroprotective functions of selegiline and rasagiline. On the other hand, clinical trials have presented only limited beneficial effects to modify disease progression. What should we do to improve the effectiveness of clinical trials? Is there any difference in clinical effectiveness between selegiline and rasagiline?

Selegiline administered as an adjunct of L-DOPA could improve akinesia and on-off oscillations [211]. In a randomized double-blind placebo-controlled phase III trial of off-patients with early PD, selegiline monotherapy (10 mg/day) was effective and tolerated for 56-week study [212]. Selegiline delayed the need for L-DOPA therapy in parkinsonian patients [213]. Several prospective, randomized, double-blind study of selegiline with or without L-DOPA showed some improvement in the UPDRS score, but the results might be ascribed to mere symptomatic effects not slowing disease progression [214].

Rasagiline exerted anti-parkinsonian activity and neuroprotection in several clinical trials, including ADAGIO study [23,24]. In the TEMPO [TVP-1012 (=rasagiline mesylate) in Early Monotherapy for Parkinson disease Outpatients] study for early PD, rasagiline (1 or 2 mg/day for 26 weeks) monotherapy favored the UPDRS score versus placebo [215]. In ADAGIO and PRESTO (Parkinson’s Rasagiline: Efficacy and Safety on the Treatment of Off) improved the UPDRS score, delayed the need for symptomatic antiparkinsonian drugs and reduced “off” time versus placebo [216,217,218]. LARGO (Lasting effects in Adjunct Therapy with Rasagiline Given One Daily) trial was designed to examine the efficacy of rasagiline as an adjunct of L-DOPA and significant reduction in L-DOPPA dose was proved with rasagiline [219]. However, long-term effects of rasagiline on the disease progress have not been fully approved [220].

Meta-analysis of randomized placebo-controlled clinical trials of rasagiline and selegiline present that a statistically significant symptomatic advantage of rasagiline in the UPDRS total scores and lowering risk for adverse events, such as dizziness, hallucination, diarrhea, and syncope [221]. In the treatment of early PD selegiline and rasagiline improved the UPDRS score at almost the same extent [222]. Another meta-analysis of clinical results of selegiline, rasagiline and safinamide in combination with L-DOPA presented that selegiline the best, followed by rasagiline [223]. Switch from selegiline to rasagiline improved motor behavior and complications, such as mood and sleep [224]. In a randomized, placebo-controlled trial for patients with parkinsonian variant of MSA, rasagiline (1 mg/day) did not show a significant improvement of Unified Multiple System Atrophy Rating Scale (UMSARS) score [225]. For disease-modifying in MSA, several clinical trials have been reported by use of L-DOPA, riluzole, amantadine, but the therapeutic benefits have not been established [226].

Selegiline and moclobemide, a reversible MAO-A inhibitor, exert antidepressant effects in clinical and preclinical studies, especially depression in PD [227]. Selegiline transdermal system was developed to bypass the first-pass metabolism and increased bioavailability to 73% from 4% of oral selegiline capsule administration [228]. In a double-blind, placebo-controlled, parallel-group study for major depression, transdermal selegiline significantly improved scores of Hamilton Depression Scale and Montogomery-Asberg Depression Rating Scale, which might be due to release of monoamine neurotransmitters by amphetamine [229]. Zydis selegiline (Zelapar) is an orally disintegrating formation of selegiline and absorbed from the buccal mucosa, increased plasma level, and decreased MAO-A PET image detected with [^11^C]clorgyline in human brain [230]. Selegiline transdermal system increases selegiline levels in the brain, inhibits MAO-A, enhances 5-HT, DA, and NA levels, and induces BDNF expression to exert antidepressant-like effects. On the other hand, rasagiline did not improve depressive symptoms, as assessed by the Beck Depression Inventory scale in a double-blind, placebo-control study in non-demented PD patients [231]. Transdermal delivery of rasagiline mesylate by use of microemulsion-based gel was proposed to increase the bioavailability for treatment of PD [232]. At present, there is no systematic trial of transdermal rasagiline in PD.

At present, delayed-start trials of rasagiline seems to present some “disease-modifying” effects. However, the results suggest the difficulty to differentiate disease-modifying effects from symptomatic effects. Selegiline transdermal system and Zydis increase the bioavailability and stability in the brain, suggesting that such drug preparations might be developed for promoting neuroprotective and antidepressant activity in PD.

## 11. Discussion and Future Aspect

Selegiline and rasagiline have presented neuroprotective effects in preclinical studies. However, disease-modification has been not proved in clinical studies, except delayed-start study of rasagiline. Limitation of clinical trials is due to unsuitable trial design, heterogeneity of parkinsonian patients, and lack of reasonable methods to follow progressing loss of dopaminergic projection from the SN to striatum. At present, available markers are the evaluation of clinical rating scale (UPDRS scale), the endpoint, and the neuroimaging studies, such as measuring DA transporter in the striatum by β-CIT SPECT, or DA neurons by [^18^F]fluorodopa PET imaging [233]. In future clinical trials, the effects should be evaluated by more sophisticated criteria in more genetically and clinically controlled subjects.

Do MAO-B inhibitors require the enzymatic activity and protein of MAOs for neuroprotection? In a mouse model of cortical infarction, selegiline suppressed ischemic injury, but did not in MAO-B-deficient mice, suggesting that MAO-B should mediate in vivo neuroprotection of selegiline [234]. MAO-B knockout reduced ^3^H-selelgiline binding to 4.0% and 2.25% of that in wild mouse striatum and cortex, and a MAO-A inhibitor clorgyline brought the binding to null, suggesting definite binding of selegiline to MAO-A, but not to other sites [235]. As discussed above, MAO-A mediates gene induction of rasagiline in neuronal cells, but the binding site in MAO-A protein to initiate signal transduction remains to be clarified. The concentrations required for upregulation of Bcl-2 and GDNF protein were quite lower than those for inhibition of MAO-B and MAO-A activity. The presence of highly sensitive binding site other than the substrate-binding or catalytic site in MAO-A was suggested. TV1022, the *S*-enantiomer of rasagiline increased Bcl-2 by activating p42/p44 MAPK in pheochromocytoma PC12 cells, which was suppressed by efaroxan, a selective I_1_ imidazoline receptor (IR) antagonist, suggesting the I_1_ IR as the binding site [236]. However, localization of I_1_ IR is localized in plasma membrane, not in MAO, whereas I2-IR is detected in MAO-A and MAO-B. Substitution of Asp328Gln in MAO-A deleted the catalytic activity but could regulate cell proliferation, de novo DNA synthesis, and enhance Bcl-2 in HEK293 cells [237]. These results suggest that there might be a highly sensitive binding site for rasagiline and selegiline in MAO to regulate survival, function, and death of neurons independently of its catalytic function.

Rasagiline prevented the mPTP initiated by PK11195 binding to TSPO, suggesting that TSPO may be a binding site for rasagiline to regulate apoptosis. TSPO is a 169 amino acid protein localized in the OMM and involved the synthesis of steroids and neurosteroids in mitochondria. It interacts with proteins implicated in the mPTP, regulates programmed cell death during neural development, synaptic formation, and axon generation, whereas TSPO in astrocytes is associated with neuroinflammation [238,239]. In the brain, TSPO is expressed mainly in microglia and reactive astrocytes and some neuronal types and PET imaging of TSPO increases corresponding to glial activation in the brain of PD, AD, amyotrophic lateral sclerosis (ALS), and HD [240]. In neurons, TSPO expression was upregulated by neuronal activity, such as physiological and psychopharmacological (amphetamine) stimuli [241]. TSPO ligands, PK11195, etifoxine [6-chloro-*N*-ethyl-4-methyl-4-phenyl-1*H*-3,1-benzoxazin-2-imine], and emapunil [XBD-173, AC-5216, *N*-benzyl-*N*-ethyl-2-(7-methyl-8-oxo-2-phenyl-purin-9-yl)acetamide] have anti-inflammatory, neuroprotective, and anxiolytic effects in clinical and preclinical studies of AD, anxiety, and depression [242,243]. Etifoxine attenuated cognitive dysfunction in LPS-treated mice by inhibition of neuroinflammation, increased synthesis of neurosteroids and prevention of apoptosis [244]. Emapunil prevented degeneration of DA neurons and motor dysfunction in MPTP-treated mouse model of PD [245]. It modulated inositol requiring kinase 1a (IREa)/X-box binding protein 1 (XBP1) unfolded protein response pathway and turned microglia to antiinflammation activation. Direct binding of rasagiline and selegiline to TSPO has not been reported and should be investigated in future.

Most preclinical studies on the neuroprotective functions have been carried out mainly in cellular and animal models prepared with oxidative stress, neurotoxins and depletion of NTFs and ATP. However, the association of PD-related genes with neuroprotection should be further investigated, using gene-transgenic animal and cell models and iPSC-derived cells. In microglia, DJ-1 downregulation with shRNA increased IL-1β and IL-6, MAO activity, and ROS/RNS, and rasagiline could reduce the proinflammatory phenotype and sensitivity to DA neurotoxicity [246]. Molecular mechanisms of rasagiline and selegiline should be investigated in such types of models to examine the effects of genetic factors in neuroprotection mechanisms. Neuroprotection is mediated by interaction between MAO-A and MAO-B localized controversially in catecholaminergic neurons and in glia and serotonergic cells. It may be interesting to clarify how MAO-A and MAO-B in different types of cells cross talk and regulate cell survival and death.

## Figures and Tables

**Figure 1 ijms-23-11059-f001:**
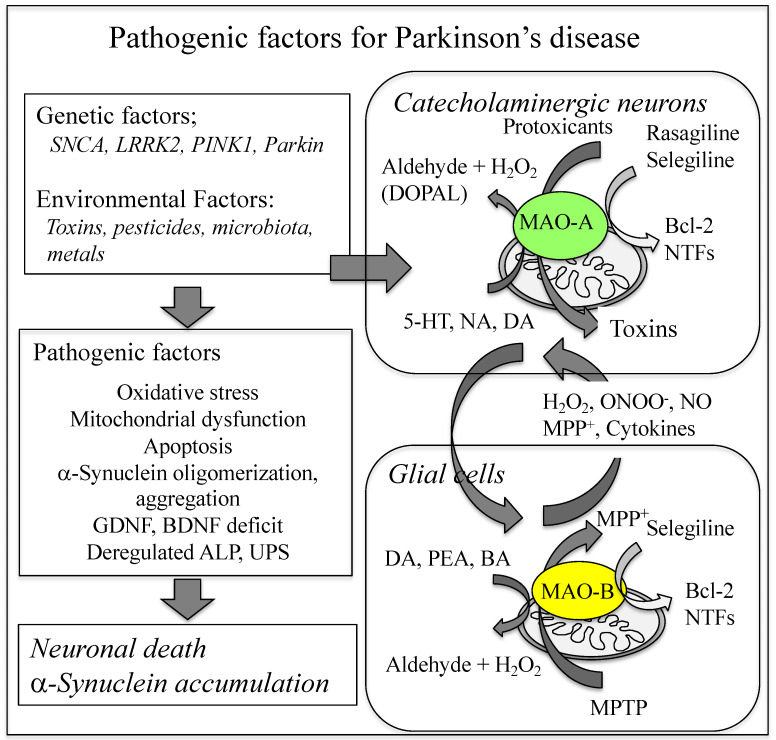
Role of MAO-A and MAO-B in the pathogenesis of PD. MAO-A in catecholaminergic neurons and MAO-B in glial cells produce ROS and neurotoxins, and induce death of DA neurons, neuroinflammation and αSyn accumulation. MAO-A mediates toxicity of endogenous dopaminergic neurotoxin, *N*-methyl(*R*)salsolinol, and gene-induction by rasagiline in dopaminergic cells. MAO-B oxidizes monoamines, produces ROS and MPP^+^ from MPTP, and mediates gene induction by selegiline in glial cells. ALP, autophagy-lysosome pathway; UPS, ubiquitin-proteasome system.

**Figure 2 ijms-23-11059-f002:**
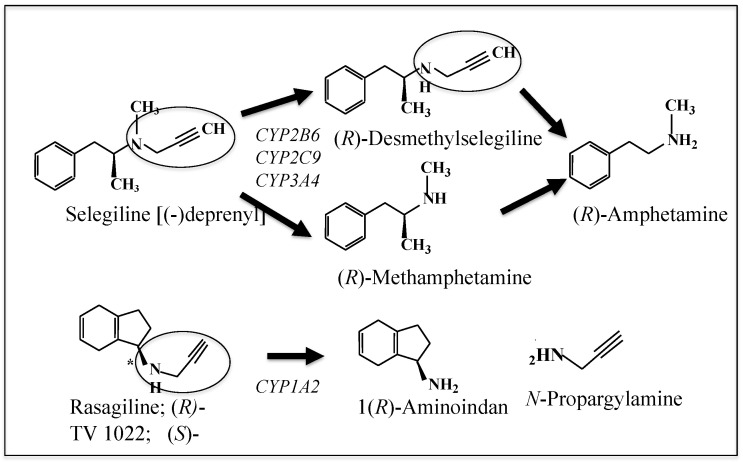
Chemical structure of selegiline and rasagiline and their metabolites. Aminoindan, a metabolite of rasagiline, has neuroprotective function, in contrast to (*R*)-methamphetamine produced from selegiline.

**Figure 3 ijms-23-11059-f003:**
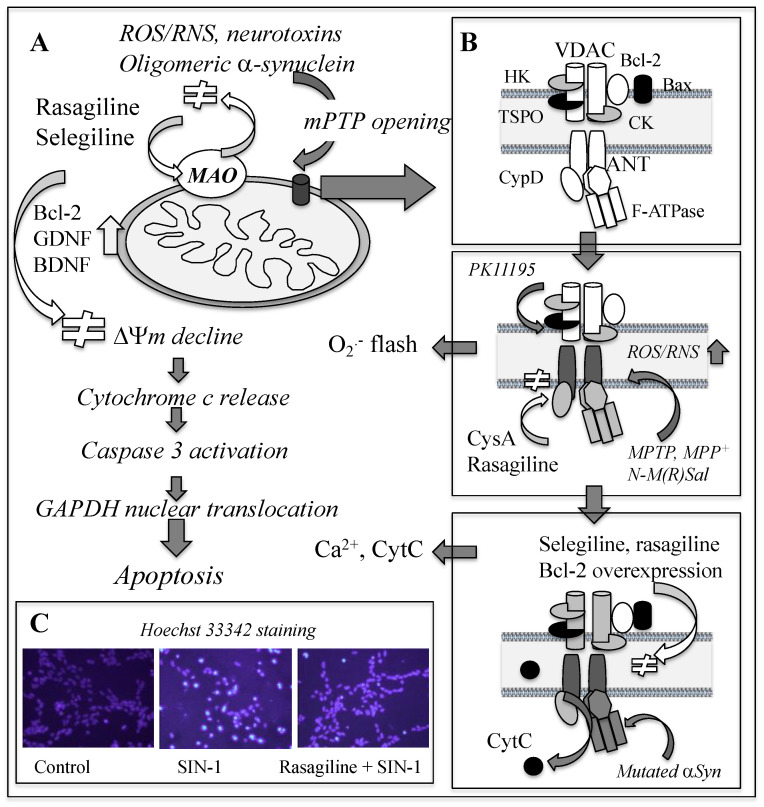
Apoptosis process and inhibition by rasagiline and selegiline. (**A**) ROS/RNS and neurotoxins induce the mPTP opening and stepwise activation of apoptosis, which rasagiline and selegiline completely prevent. Mutated αSyn binds to and inhibits ATP synthase and opens the mPTP. (**B**) Modulation of the mPTP formation by rasagiline and selegiline. PK11195, a TSPO ligand, opens a pore composed of ANT and CypD at the IMM, shown as superoxide flash, which CysA and rasagiline prevent. Then, the mPTP open with Ca^2+^ release occurs, which Bcl-2 and selegiline suppress. (**C**) Nuclear condensation induced by ONOO^−^-generating SI*N*-1 in SH-SY5Y cells. Rasagiline inhibits the nuclear condensation visualized by Hoechst 33342 staining.

**Figure 4 ijms-23-11059-f004:**
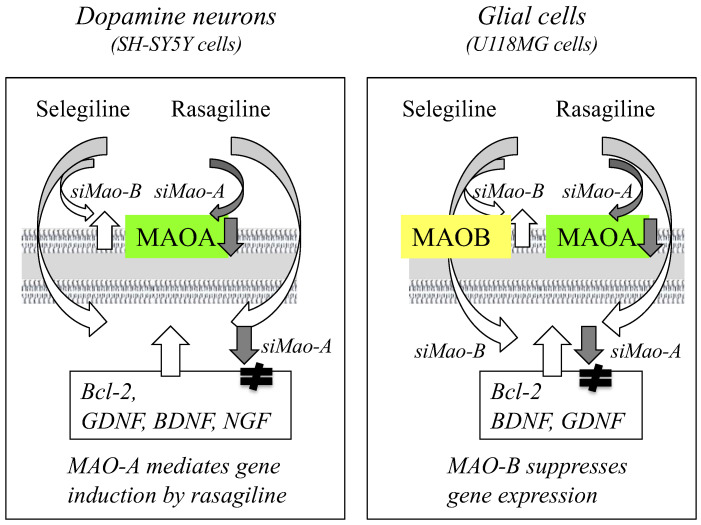
MAO-A and MAO-B are associated with gene induction by rasagiline and selegiline. Knockdown of MAO-A with siRNA downregulated gene expression by rasagiline in neuroblastoma SH-SY5Y cells, whereas MAO-B knockdown upregulated gene induction by selegiline in glioblastoma U118MG cells.

**Figure 5 ijms-23-11059-f005:**
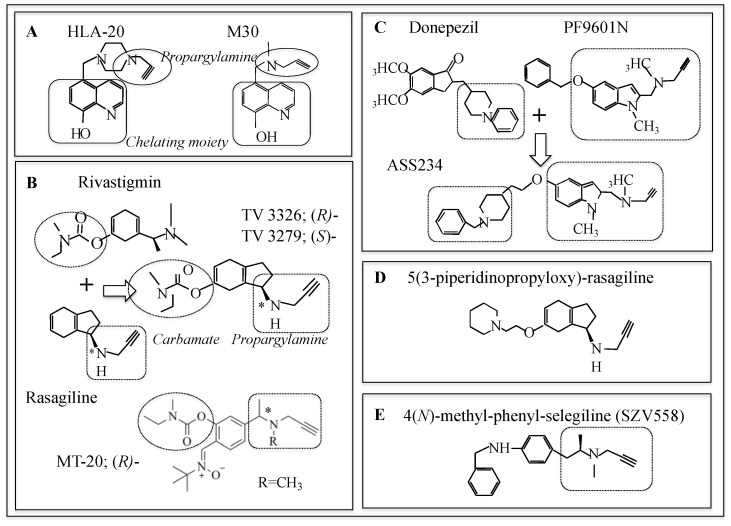
Chemical structure of propargylamine derivatives. (**A**) 8-Hydroxyquinone derivatives of rasagiline (HLA-20, M30) inhibit MAO-B and chelate metals. (**B**) Propargyl derivatives of rivastigmine (TV3326, TV3279) inhibit ChE and MAO-B with a carbamate moiety and exert neuroprotection by a propargylamine moiety. (**C**) Propargylamine derivatives of donepezil. (**D**) Indanone derivatives of rasagiline are H3 receptor agonists, inhibit MAO, and improve cognitive functions. (**E**) *N*-Methyl-phenyl-selegiline inhibits MAO-B, chelates copper (II), iron (II), and zinc (II), and inhibits Aβ_1–42_ aggregates.

**Table 1 ijms-23-11059-t001:** Pharmacokinetic data of selegiline and rasagiline.

Parameters	Selegiline [61]	Rasagiline [62]
C_max_ (µg/L)	2.20 ± 1.17	17.55 ± 3.51, PD, 14.9 ± 10.5
T_max_ (h)	0.90 ± 0.38	0.4 ± 0.2, PD, 0.5 ± 0.7
T_1/2_ (h)	1.20 ± 0.56	2.06 ± 1.14, PD, *nr*
AUC (µg.h/L)	3.75 ± 0.28	20.02 ± 4.81, PD, 23.5 ± 10.5
CL/F (L/min)	59.4 ± 43.7	33.6 ± 7.8, PD, *nr*
Bioavailability (%)	<10	36

Selegiline (10 mg/day) was once only orally administered to healthy human subjects [61], and rasagiline (2 mg/day) to control and PD patients as an adjunct to L-DOPA UC, area under the plasma concentration-time curve; CL/F, apparent total body clearance; C_max_, maximal plasma content; T_max,_ time to reach the maximal level; T_1/2_, half-life time. *nr*, not reported; PD, parkinsonian patients.

## Data Availability

All the data are available within the article.

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
