# Peer review of "Neuroprotective Function of Rasagiline and Selegiline, Inhibitors of Type B Monoamine Oxidase, and Role of Monoamine Oxidases in Synucleinopathies"

_ijms, 2022, doi:10.3390/ijms231911059_

Round 1
Reviewer 1 Report
This is an interesting review about the role of MAO-B inhibitors for the synucleinopathies. I believe its application for Parkinson’s disease was already well known, but this review amplifies its application to the entire spectrum of synucleinopathies.
However, authors should highlight which are the gaps in the current knowledge and add what is the novel contribution of the present review into the manuscript.
It would be great to add a sentence to introduce about what is the paragraph about at the beginning of each subsection. I would also encourage to add a take-home message after each section. The paragraph at the end of the introduction is a good summary of the content of the review, however in the further sections, it would be great to add a short take-home message, so that the reader can follow it in a more understandable way. Furthermore, the use of connectors for some sentences would be very much appreciated. I would suggest to have the manuscript revised by a native/fluent English speaker.
Here are some of my suggestions (feel free to use them if you like them):
- Line 35: “aSyn is a protein of 140 amino acids, highly expressed at the presynaptic compartment, which regulates synaptic maintenance, DA neurotransmission, mitochondrial homeostasis, and proteasome function.”
- Line 50: “PD is the second most common ageing-related neurodegenerative disease after AD.”
- Line 133-134: “MAO-A is mainly expressed in catecholaminergic neurons and most abundant in the locus coeruleus...”.
- Line 138: “MAO-A levels in the brain can already be determined before birth, contributing to embryonic development of brain architecture and affecting postnatal behavior and emotional state.”
Minor concerns:
I would appreciate if the authors could differentiate between in vitro, in vivo or human postmortem studies when citing them, as sometimes it’s not completely clear (i.e. line 308: PD brains (humans, animal models?)).
I also consider that the manuscript would benefit a lot if a better design of the visuals could be provided.
Author Response
Thank you for your comments and advice to improve my paper. I totally revised my paper and changed the order of sentences from Session 5 to Session 9. In the submitted version, the mechanisms of neuroprotection by rasagiline and selegiline was emphasized too more intensively than the role of MAOs. I added several results on the role of MAO in the neurodegeneration and MAO-B inhibitor-dependent neuroprotection. I added several references as another referee commented. I high-lighted the corrected and newly added parts in yellow.
I revised Introduction and text to point out the role of type A monoamine oxidase in the neurodegeneration of synucleinopathies and the neuroprotection by type B monoamine oxidase mainly based on our data to emphasize novel issues in my paper.
According to your comments, I inserted sentences at the front and end of each session to clarify the major subjects to discuss in each session, and to present important or interesting points of presented data.
I corrected the sentences (Lines 35, 50, 133-134, 138) according to your advice and show them in yellow.
I tried to present the types of experiments for the presented results.
This IJMS commonly does not include many figures and I limited the number of figures.
Reviewer 2 Report
Makoto Naoi and co-authors presented a literature review about the neuroprotective function of rasagiline and selegiline, and the role of monoamine oxidases in synucleinopathies. The authors discuss possible reasons why preclinical studies have presented neuroprotection of rasagiline and selegiline but clinical trials have scarcely presented beneficial effects. Also, the authors consider possible strategies for improve clinical trials to achieve disease-modification in synucleinopathies.
Overall, the article is well written and only some minor corrections required:
Point 1: The article is missing an abstract.
Point 2: In some cases, the paper does not contain references to the data presented (e.g., see lines 120-125, 138-142, 197-205).
Point 3: Delete the extra dot after "rasagiline.," (line 109).
Point 4: The abbreviation mPTP is first mentioned in line 293 but is only spelled out in line 297.
Point 5: There are extra dots before the names of some sections and subsections (see lines 321, 332, 542, 629, 691).
Point 6: Insert spaces after: "Author's contribution:" (line 751), "Funding:" (line 753), "Conflict of interest:" (line 755).
Point 7: References incorrectly formatted. Additionally, authors should include DOI for all references where available.
Point 8: Since the article contains a large number of abbreviations, I recommend presenting a separate abbreviation list.
Author Response
Thank you for your comments and advice to improve my paper. I totally revised my paper and changed the order of sentences from Session 5 to Session 9. In the submitted version, the mechanisms of neuroprotection by rasagiline and selegiline was emphasized too more intensively than the role of MAOs. I revised Introduction and text to point out the role of type A monoamine oxidase in the neurodegeneration of synucleinopathies and the neuroprotection by type B monoamine oxidase mainly based on our data. Major revised parts are shown in yellow.
According to the comment of another referee, I inserted sentences at the front and end of each session to clarify the major subjects to discuss in each session, and to present important or interesting points of presented data.
I re-arranged order of references to fit the changed order in the text. I added several references as you commented. I heighted the corrected and newly added parts in yellow.
Answer to your comments.
Point 1: Sorry to the absence of the Abstract. For submission, I used the older template for IJMS and the abstract was deleted during download process for submission. There occurred several deforms in my submitted version, which I correct by the revision.
Point 2: I cited references for Lines 120-125, 138-142, 197-205. Added references are high-lighted in yellow.
Point 3 and 5: I deleted dot, line 109. 321, 332, 542. 529, 192.
Point 4: “mPTP” was defined.
Point 6: I corrected (Lines 751, 753, 755).
Point 7: References were formatted in the style used in this journal. DOI is not included in IJMS.
Point 8: I like to show a list Abbreviation. But this journal does not present. I asked to editor if the list can be included.
Round 2
Reviewer 1 Report
The revised manuscript has addressed many of the points that I raised, and I appreciate the authors willingness to revise their paper. I believe that the revised paper is clearer and provides much of the documentation I requested.
Notwithstanding, I would suggest to the authors to have a final revision for fine/minor spelling mistakes.